# Comparison on Protein Bioaccessibility of Soymilk Gels Induced by Glucono-δ-Lactone and Lactic Acid Bacteria

**DOI:** 10.3390/molecules27196202

**Published:** 2022-09-21

**Authors:** Tianran Hui, Ting Tang, Xuan Gu, Zhen Yuan, Guangliang Xing

**Affiliations:** 1School of Biology and Food Engineering, Changshu Institute of Technology, Changshu 215500, China; 2Department of Biological and Environmental Sciences, Troy University, Troy, AL 36082, USA

**Keywords:** soy protein, in vitro digestion, bioaccessibility, lactic acid bacteria, glucono-δ-lactone

## Abstract

In this study, the protein bioaccessibility of soymilk gels produced by the addition of glu-cono-δ-lactone (GDL) and fermentation with lactic acid bacteria (LAB) was examined using an in vitro gastrointestinal simulated digestion model. The in vitro protein digestibility, soluble protein content, free amino acids contents, degree of hydrolysis, electrophoretic patterns, and peptide content were measured. The results suggested that acid-induced soymilk gel generated by GDL (SG) showed considerably reduced in vitro protein digestibility of 75.33 ± 1.00% compared to the soymilk gel induced by LAB (SL) of 80.57 ± 1.53% (*p* < 0.05). During the gastric digestion stage, dramatically higher (*p* < 0.05) soluble protein contents were observed in the SG (4.79–5.05 mg/mL) than that of SL (4.31–4.35 mg/mL). However, during the later intestinal digestion phase, the results were the opposite. At the end of the gastrointestinal digestion phase, the content of small peptides was not significantly different (*p* > 0.05) between the SL (2.15 ± 0.03 mg/mL) and SG (2.17 ± 0.01 mg/mL), but SL showed higher content of free amino acids (20.637 g/L) than that of SG (19.851 g/L). In general, soymilk gel induced by LAB had a higher protein bioaccessibility than the soymilk gel coagulated by GDL.

## 1. Introduction

Soybean (*Glycine max*) has been widely cultivated around the world and has occupied a very important position in the agricultural industry due to its high nutritional value. Soybean curd, commonly known as tofu, is a delectable dish made by coagulating cooked soymilk with or without pressing soy whey [1]. In the soybean curd production procedures, different coagulants significantly affect the quality and yield of tofu [2]. Traditionally, common coagulants can be divided into salt coagulants (for instance, magnesium and calcium salts) and acid coagulants (such as glucono-δ-lactone [GDL]) [3]. Recently, soybean curd with probiotic qualities and improved nutritional values has been produced by using lactic acid bacteria (LAB) [4,5,6]. Fermentation of soymilk with LAB has been reported to enhance the bioactive components, including γ-aminobutyric acid and aglycone isoflavones, which are responsible for health benefits such as antioxidant, anticancer, antihypertensive, antidiabetic, and hypocholesterolemic effects [7]. Several LABs could produce proteolytic enzymes during the fermentation of soymilk, which generates bioactive peptides [8,9]. These peptides could improve the functionality of fermented food and can be used as a biological substitute for a variety of synthetic drugs [10]. Moreover, exopolysaccharide (EPS)-producing LAB could improve the consistency and rheology of fermented soymilk products by modifying fluid flow characteristics [11]. The lubrication properties of soymilk gels could also be efficiently improved with an EPS-producing LAB culture compared with GDL-induced gels [12]. The regular consumption of lactic acid-fermented foods will benefit consumers nutritionally and serve as an immunity booster against diseases and infections [13].

The diverse quality and texture characteristics of tofu induced by different coagulants are mostly attributable to the presence of various forces to stabilize the gel network. Gelation mechanisms of tofu via the coagulants mentioned above have been well documented [14,15]. Hydrogen bonds, hydrophobic interactions, van der Waals forces, and disulfide bonds had been found to play important roles in the formation of acid-induced networks in soymilk gels [14]. The differences in the soymilk gelation process caused by LAB or GDL have been researched [16]; however, the nutritional characteristics of soy protein between these two acid-induced soymilk gels are unknown.

In vitro digestion involves the digestion of substances in a controlled environment based on the physiological conditions of the gastrointestinal tract. During this process, the pH value, concentration of digestive enzymes, temperature, and peristalsis of the gastrointestinal tract of the organism can be simulated, which can represent the changes in substances in the organism to a certain extent [17]. The ingested nutrients are bioaccessible, meaning that they can be absorbed and utilized by the host organism [18]. Compared to in vivo digestion, in vitro simulated digestion has the advantages of simplicity, convenience, rapidity, low cost, and reproducibility, which has been widely used in the field of food or drugs, such as studying the bioavailability of different components in food, determining the stability of food during digestion, and advancing the slow release of drugs [19].

Therefore, this work aimed was to assess the protein bioaccessibility of two soymilk curds prepared by GDL or fermented with LAB, using an in vitro gastrointestinal simulated digestion (GIS) model. The in vitro protein digestibility, degree of hydrolysis (DH), peptide content, protein degradation profiles (sodium dodecyl sulfate-polyacrylamide gel electrophoresis (SDS-PAGE)), soluble protein content, and free amino acids contents were investigated. The findings of this study contribute to understanding the difference in protein bioaccessibility between the acid-induced aggregation of soymilk particles using GDL and LAB. It will also help provide fundamental knowledge for future applications of LAB in the acid-induced soymilk gel manufacturing industry with high nutrient-releasing properties.

## 2. Results and Discussion

### 2.1. Proximate Analysis

Proximate analysis was carried out to examine the chemical compositions of soymilk gels. The results showed that the soymilk gel produced by GDL (SG) possessed 3.15 ± 0.08 g of protein, 1.72 ± 0.11 g of lipid, 0.96 ± 0.05 g of carbohydrate, 93.81 ± 0.76 g of moisture, and 0.34 ± 0.03 g of ash contents per 100 g (wet basis), respectively. For the soymilk gel induced by LAB (SL), protein, lipid, carbohydrate, moisture, and ash contents were 3.17 ± 0.05 g, 1.69 ± 0.08 g, 0.92 ± 0.07 g, 93.77 ± 0.85 g, and 0.38 ± 0.08 g per 100 g (wet basis), respectively.

### 2.2. Protein Degradation during GIS Digestion

The changes in soluble protein concentration in soymilk (control), soymilk gel produced by GDL (SG), and soymilk gel induced by LAB (SL) samples are illustrated in Figure 1 during various phases of gastrointestinal digestion. Grygorczyk and Corredig [16] reported that the gelation modes of soymilk induced by the acidification of LAB and GDL were similar, but the pH value of the gel point was different. In the present study, within 30 min, the addition of 0.3% (*w*/*w*) GDL acidified soymilk to pH 5.6. Previous research has shown that lactic acid fermentation terminated at different pH values can alter soy protein digestibility [20,21]. Therefore, the pH of soymilk fermented with LAB was also terminated at approximately 5.6 after about 240 min at 42 °C.

As shown in Figure 1, the amount of soluble protein in soymilk (7.52 mg/mL) was substantially higher (*p* < 0.05) than that in SG (0.93 mg/mL) and SL (0.84 mg/mL) and before digestion (P0 stage). This indicated that most of the proteins existed in a free state in soymilk, while in the SG and SL gels, most of the proteins remained in the gel matrix. The mechanisms of acid-induced gelation of soymilk gel, in general, consist of two steps: heat treatment-induced protein denaturation followed by acid addition to generate protein coagulation [22]. In the present study, the hydrophobic groups of soy protein in soymilk were exposed after heat treatment (100 °C, 5 min), and then soluble protein aggregates formed. The release of protons (produced by GDL hydrolysis or generated by LAB fermentation) neutralized the negative charges on the surface of soluble aggregates in cooked soymilk, the repulsion among soy proteins disappeared and aggregation occurred [23]. After high-speed centrifugation (10,000× *g* r/min for 15 min), the proteins remained in the soymilk gels and precipitated to the bottom of the centrifuge tube, while few proteins existed in the supernatant. Therefore, the protein solubility of SG and SL gels before digestion (P0) is expected to be much lower than that of soymilk. Similar results were observed for all investigated samples after buccal digestion (P1) compared to the P0 stage, which suggested that the α-Amylase had little impact on the amount of soluble protein. These results corroborate findings from a previous study [24], which compared the protein bioaccessibility of soymilk and soymilk curds generated by fermentation with different LAB strains, at both P0 and P1 stages, where much lower soluble protein contents were observed in the soymilk curds compared to soymilk.

Both SG and SL gels displayed a similar pattern of soluble protein concentration during the gastrointestinal digestion stages (P2 and P3), which had an apparent increase compared to those of the P0 and P1 phases. This is mainly because of the continuous enzymatic hydrolysis of pepsin and pancreatin; large proteins were degraded to form small soluble molecules in the digestive process [25]. Regarding the gastric digestion stage (P2), the soluble protein concentration was noticeably higher in SG (4.79–5.05 mg/mL) than SL (4.31–4.35 mg/mL) (*p* < 0.05). However, the results were the exact opposite during the later P3 stage, with SL having a significantly larger soluble protein level (2.62–3.07 mg/mL) than SG (1.77–2.65 mg/mL) (*p* < 0.05). Notably, soluble protein concentration was lower in all tested samples during the intestinal digestion (P3) phases than it was during the gastric digestion (P2) phases. This is probably because proteolysis induced by pepsin and/or pancreatin could produce a more heterogeneous protein profile and smaller-molecular- weight peptides/amino acids that were unable to be detected by the Bradford assay [26].

### 2.3. The DH and Protein Digestibility

The DH refers to the proportion of peptide bonds in protein molecules that are broken due to enzymatic hydrolysis to the total peptide bonds in protein molecules [27]. During in vitro GIS digestion, soy protein is hydrolyzed by proteases to release various forms of small peptides and free amino groups (-NH_2_). The DH is determined by measuring the amount of free -NH_2_ released during digestion by the OPA method [28]. During the digestion phase, each hydrolysis of a peptide bond releases a free -NH_2_, and the free -NH_2_ reacts with OPA to form a yellow complex, which can be measured by a UV spectrophotometer (UV-1800, Mapada Instruments Co., Ltd., Shanghai, China) at 340 nm. Thus, the number of broken peptide bonds can be determined by the number of newly formed terminal-NH_2_ groups after hydrolysis, and the DH can be indirectly obtained by the absorbance value, the larger the absorbance value, the greater the degree of hydrolysis.

The DH as a function of digestion time is shown in Figure 2. Before digestion (P0), both SG and SL samples had a relatively low DH compared to that of the soymilk (control). After the buccal digestion (P1), the DH of the soymilk gel samples was still low, indicating that there was no obvious hydrolysis of soy protein during this period. This could be attributed to the short digestion time (3 min) and the weak ability of α-Amylase to hydrolyze soy proteins. Because the number of cleaved peptide bonds in soy protein of the soymilk gel samples before digestion (P0) and after buccal digestion (P1) was very small, very few free -NH_2_ groups were released. According to the formulas in Section 3.6, the degree of hydrolysis in SG and SL were negative values at the P0 and P1 stages. These findings are supported by Rui et al. [24], who reported that the soymilk curds fermented by LAB had negative DH values at the initial digestion phases of P0 and P1 compared to soymilk.

However, in the following gastric digestion stage, the DH of both soymilk gel samples increased sharply, and positive values were obtained, suggesting that extensive protein hydrolysis occurred. At the end of gastric digestion (P2-60), the DH of soymilk, SG, and SL was 14.42%, 12.34%, and 10.77%, respectively, with significant differences (*p* < 0.05). Many factors contributed to this result, including the acidic environment during gastric digestion, the presence of pepsin, the continuous mechanical vibration, and the sufficient digestion time [29]. During the intestinal digestion phase (P3), the DH of the SG and SL gels increased from 12.16% to 13.64% and from 12.85% to 14.57%, respectively. That is because, in the presence of pancreatin and bile salts, a large number of proteins were further hydrolyzed into various small peptides, causing the peptide bonds to break in large quantities, and a continuous upward trend in DH was observed. Throughout gastrointestinal digestion, the DH of soymilk showed a steady increase from 12.52% (P0) to 16.75% (P3-120), indicating that digestive enzymes promoted the hydrolysis of soy protein. These results are consistent with previous studies. For example, Yang et al. [30] also reported very low DH values of soymilk gels induced by *Lactobacillus casei* (0.26%), GDL (0.01%), and citric acid (0.1%) before the in vitro digestion phase. However, after gastric and intestinal digestion, the DH of these three gels increased to more than 10%. Similar results were also observed for the potato protein isolate with different treatments, and all samples showed a drastic increase in DH from gastric to their respective intestinal points due to efficient activity by trypsin and chymotrypsin [31].

In general, the rapid hydrolysis of proteins and the increased number of low molecular weight peptides were responsible for the continuous increase in the degree of hydrolysis [32]. The trend of the hydrolysis degree from the P0 stage to the P3 stage revealed that the digestive enzymes, time, and the change in pH affected the hydrolysis degree.

The in vitro protein digestibility of soymilk and both soymilk gels (SG and SL) is shown in Table 1. We found that the soymilk had much higher (*p* < 0.05) in vitro protein digestibility than both soymilk gels. Moreover, the in vitro protein digestibility of SL (80.57 ± 1.53%) was observably higher (*p* < 0.05) than that of SG (75.33 ± 1.00%), indicating that SL was more susceptible to digestive enzymes. It has been previously shown that when bacterial acidification occurs, the soy proteins approach the isoelectric point quite slowly. Therefore, soy proteins have more time to reorganize and interact with one another [16]. Bacterial acidification might result in lower interacting forces between soymilk proteins and accelerated proteolysis during in vitro protein digestion.

### 2.4. Bioaccessible Peptides

Figure 3 displays the peptide (<10 kDa) contents of the two soymilk gels and the control. Before digestion (P0) or after the buccal digestion (P1), all samples showed very low peptide content (approximately 1.0 mg/mL). This indicated that both salivary amylase and chewing failed to hydrolyze soy proteins. However, due to the gastric and duodenal fluids containing digestive enzymes, the peptide content rose as GIS digestion progressed. This observation is consistent with the results obtained by other authors [29]. None of the tested samples showed significant differences after gastric digestion (P2-60). However, at P3-120, significantly higher values (*p* < 0.05) were obtained from soymilk digest than soymilk gels, indicating that the acid-induced soymilk gels were both less susceptible to digestive enzymes than soymilk samples. The content of small peptides was not significantly different (*p* > 0.05) between the SL (2.15 ± 0.03 mg/mL) and SG (2.17 ± 0.01 mg/mL) at the end of the gastrointestinal digestion phase. Similar results can be found in Hall et al. [33], who stated that the peptide contents (<10 kDa) in both lentil and faba bean protein concentrate after intestinal digestion were significantly higher than those after gastric digestion. It has been reported that most of the biologically active peptides from legume proteins are small, especially those with a short length of 2–10 amino acids [34]. Our findings imply that SG and SL gels had an equivalent possibility of releasing bioactive peptides from acid-induced curds during in vitro GIS digestion.

### 2.5. SDS-PAGE

To comprehend the protein profile changes, further electrophoresis was carried out (Figure 4). The main soy protein constituents in the control were visible in both the P0 and P1 phases, as shown in Figure 4a. Based on their estimated molecular masses of 83.4, 74.6, 49.0, 39.7, 35.6, and 20.1 kDa, respectively, these constituents were hypothesized to be the 7S α′, 7S α, and 7S β subunits (bands numbered as 1, 2, and 3, respectively), 11S A3 subunit (band numbered as 4), 11S acidic subunits (bands numbered as 5), and 11S basic subunits (bands numbered as 6) [35]. In contrast, all these bands were reduced in the P1 stage compared to the P0 stage because of the dilution effect of the simulated saliva. Both SG and SL gels showed fewer bands in the P0 stage than soymilk, and only four major bands (numbered 7–10, with corresponding molecular masses of 62.5, 42.8, 27.1, and 17.6 kDa) were observed, indicating that the majority of proteins were confined to the soymilk gel. These results are in agreement with the soluble protein content at a low level (<1.0 mg/mL) initially obtained and shown in Figure 1. A previous study also stated that only a few soy protein bands could be observed on the electrophoretic protein pattern of soymilk gels induced by LAB fermentation, because of the high interacting forces between soymilk proteins, thus hindering the release of those proteins into the soluble fraction [21]. It can be observed that 7S α′, 7S α, and 11S acidic subunits were absent in the soluble fractions of both SG and SL gels, whereas the molecular masses of bands 8 and 10 were close to those of bands 4 and 6, from which it could be determined that bands 8 and 10 correspond to the 11S A3 subunit and 11S basic subunit, respectively.

In all samples, further gastric digestion resulted in the elimination of 7S and 11S globulins, and smeared bands (15–20 kDa) were generated (P2, Figure 4a,b). This indicates that the organized protein network began to disintegrate during this phase, and the majority of the protein macromolecules experienced a fast breakdown. Other researchers had similar observations. Lou et al. [36] reported that both CaSO_4_ and GDL tofu shifted toward low-molecular proteins based on the temporal evolution of SDS-PAGE bands at the end of gastric digestion. Xu et al. [37] stated that the macromolecular proteins of soymilk were rapidly digested after gastric digestion, leaving only small molecules at the lower part (<20 kDa). In the intestinal digestion phase (P3), the SG and SL digests displayed electrophoretic profiles that resembled those of soymilk in terms of molecular mass (as denoted by arrows), but higher intensities were observed in the SL digest bands. This suggests that there were greater interaction forces between soymilk proteins during GDL-induced protein coagulation, which prevented the release of protein into the soluble fraction. This is in agreement with previous digestibility studies on lentil and faba bean protein concentrates that were hydrolyzed to small peptides (<3 kDa) after intestinal digestion [33]. These findings are consistent with prior results in terms of in vitro protein digestibility and the amount of soluble protein.

### 2.6. Amino Acids Analysis

At the end of the in vitro GIS digestion phase (P3-120), the concentration of free amino acids in the soymilk and both soymilk gels were determined, and the findings are reported in Table 2. There were 17 amino acids found, including 8 essential amino acids (threonine (Thr), valine (Val), methionine (Met), isoleucine (Ile), leucine (Leu), phenylalanine (Phe), lysine (Lys), and tryptophan (Trp)). The total essential amino acid content of all the tested samples was similar. In particular, the total free amino acids content of the SL digest was 20.637 g/L, which was approximately 5% greater than that of the SG digest. Lys (5.550), Tyr (4.638), Trp (3.961), Leu (2.476), and Phe (1.764) were the most prevalent amino acids released (in grams per liter) in SL, accounting for 89.1% of total free amino acids.

## 3. Materials and Methods

### 3.1. Materials

Soybean seeds were bought from Shanggong Food Co., Ltd., Zaozhuang, Shandong, China. Commercial lyophilized LAB culture (Angel Yeast Co., Ltd., Yichang, Hubei, China) was used to ferment soymilk. The most suitable temperature for growth is 42 °C, and the culture contains two strains (*Lactobacillus delbrueckii* ssp. *bulgaricus* and *Streptococcus thermophilus*). Glucono-δ-lactone (GDL) was also purchased from Angel Yeast Co., Ltd., Yichang, Hubei, China. Bromophenol blue, bovine serum albumin, Coomassie Brilliant Blue R250, and Coomassie Brilliant Blue G250 were obtained from Feijing Biotechnology Co., Ltd., Fuzhou, Fujian, China. Sodium dodecyl sulfate (SDS), tris base, glycine, sodium tetraborate decahydrate, and *o*-phthalaldehyde (OPA) were purchased from Yien Chemical Technology Co., Ltd., Shanghai, China. L-serine, casein tryptone, pancreatin (P7340), pepsin 1:3000 (P8390), pig bile salts (G8310), α-Amylase (G8290), and β-mercaptoethanol were obtained from Solarbio Science and Technology Co., Ltd., Beijing, China. Trypsin 1:200 (porcine pancreas, S10034), Chymotrypsin (S10001), and peptidase (S31740) were brought from Yuanye Biotechnology Co., Ltd., Shanghai, China. Pre-stained molecular mass standard protein markers (15–130 kDa) and the SDS-PAGE gel rapid preparation kit were purchased from Ranjeck Technology Co., Ltd., Beijing, China.

### 3.2. Preparation of Soymilk Gels

The process of preparing soymilk gels is shown in Figure 5. Soybean seeds (200 g) were soaked in 600 mL of distilled water at 25 °C for 12 h. The swollen soybeans were then ground using a soymilk blender (L18-Y915S, Joyoung Co., Ltd., Hangzhou, China) with another 1400 mL of distilled water. Subsequently, filtration was performed using a 160-mesh sterile gauze to collect raw soymilk (approximately 1580 mL), and then heated to 100 °C for 5 min. About 1500 mL of the cooked soymilk was taken and divided equally into three volume parts. The first part was cooled to approximately 85 °C and mixed with 0.3% (*w*/*v*) GDL. The mixture was transferred to a beaker and then heated at 85 °C in a water bath. The pH value was recorded as 5.6 after heating for 30 min (using a pH meter FE20, Mettler Toledo Technology (China) Co., Ltd., Shanghai, China), and the soymilk gel was obtained with the addition of GDL (SG). The second part of the cooked soymilk was cooled to approximately 42 °C in a water bath (kept at 42 °C) and inoculated with approximately 10^7^ culture-forming units (CFU)/mL LAB strains. Before use, commercial lyophilized bacteria were suspended in sterile water. The pH values were noted hourly throughout the incubation period, and fermentation was terminated when the pH value approached 5.6 (approximately 4 h later) and the fermented soymilk gel (SL) was obtained. The third part of the cooked soymilk was set as the control.

### 3.3. Proximate Analysis of Soymilk Gels

Soymilk gels were analyzed for fat, moisture, and ash in accordance with the American Association of Cereal Chemists [38]. Protein was determined based on the Kjeldahl method, employing a nitrogen-to-protein conversion factor of 6.25. Determination was conducted in triplicates.

### 3.4. In Vitro Gastrointestinal Simulated Digestion

The in vitro gastrointestinal simulated digestion (GIS) was carried out as the method adapted from Hui and Xing (2022) [39]. The whole in vitro GIS digestion steps are shown in Figure 6. Firstly, both the soymilk gel samples (SG and SL) were crushed with a hand-held homogenizer to simulate the chewing process, and a 5 g sample was taken as the undigested sample, numbered P0. Then, 60 g of the crushed soymilk gel samples or control were collected, adding 24 mL of salivary amylase solution (0.2 mg α-Amylase/mL, dissolved in 20 mmol/L phosphate buffer, pH 7.0), and the mixture was shaken at a constant temperature of 37 °C on an electro-heating standing-temperature cultivator (DHP-9052, Yiheng Scientific Instrument Co., Ltd., Shanghai, China) at 55 r/min for 3 min, and 7 g of the sample was taken out as the sample after saliva digestion, numbered P1. Subsequently, 4 mol/L HCl was used to adjust the pH of the system to 2.0 ± 0.02, and then 33 mL of pepsin juice (3.2 mg pepsin/mL, dissolved in 0.1 mol/L HCl) was added. The rotation speed remained unchanged (55 r/min), and the gastric digestion time was 1 h. At the 5 min and 60 min of the gastric digestion phase, 10 g of samples were collected and numbered as P2-5 and P2-60, respectively. After gastric digestion, 4 mol/L NaOH was used to adjust the pH of the system to 7.0 ± 0.02, and 18 mL of pancreatic juice (0.4 mg pancreatin/mL, dissolved in 20 mmol/L phosphate buffer, pH 7.0) and 18 mL of bile solution (0.4 mg bile salts/mL, dissolved in 20 mmol/L phosphate buffer, pH 7.0) were added to simulate intestinal digestion for 2 h at 150 r/min. During this period, 14 g of samples were collected at 30 min and 120 min, respectively, and numbered as P3-30 and P3-120. To confirm that the protein content remained consistent across all samples, the digested samples collected at each digestion step were filled to 14 mL with distilled water.

Hence, aliquots were taken at the following intervals: without digestion (P0); after buccal digestion (P1); gastric digestion at 5 min (P2-5) and 60 min (P2-60); intestinal digestion at 30 min (P3-30) and 120 min (P3-120). To terminate the enzymatic hydrolysis, all collected samples were immersed in boiling water immediately for 3 min and then centrifuged at 10,000× *g* r/min for 15 min. The supernatants were collected and kept at 4 °C until further analysis.

### 3.5. Soluble Protein Content Determination

The Bradford assay [40] was used to determine the amount of total soluble protein present in the supernatant of the digested and undigested samples indicated in Section 3.4 One milliliter of the supernatant was mixed with 5 mL of Coomassie Brilliant Blue G-250 solution, and after 5 min of reaction at room temperature, the absorbance at 595 nm was measured on an ultraviolet-visible spectrophotometer (UV-1800, Mapada Instruments Co., Ltd., Shanghai, China). Bovine serum albumin was used as a standard. Three replicates were conducted for each sample.

### 3.6. The Degree of Hydrolysis and In Vitro Protein Digestibility

An improved o-phthaldialdehyde (OPA) method was applied to determine the degree of hydrolysis (DH) as previously described by Nielsen et al. [28], the calculation formulas are as follows. L-serine (Solarbio Science and Technology Co., Ltd., Beijing, China.) was used as the standard.
(1)$$W_{\mathrm{serine}-{\mathrm{NH}}_{2}}=\frac{{\mathrm{OD}}_{\mathrm{sample}}-{\mathrm{OD}}_{\mathrm{blank}}}{{\mathrm{OD}}_{\mathrm{stand}}-{\mathrm{OD}}_{\mathrm{blank}}}\times 0.9515\times \frac{V\times N}{X\times P}$$
(2)$$h=\frac{W_{\mathrm{serine}-{\mathrm{NH}}_{2}}-\beta }{\alpha }$$
(3)$$\mathrm{DH}=\frac{h}{h_{\mathrm{tot}}}$$
where W_serine-NH_2__ = Millimolar equivalent serine-NH_2_/g protein; OD_sample_, OD_blank_ and OD_stand_ = absorption values of sample tube, blank tube and standard tube at 340 nm, respectively; 0.9515 (mmol/L) = concentration of serine standard solution; N is the dilution ratio of digestive fluid; V is the sample volume in liter (L); X = g sample; P = protein % in sample; α and β are estimated to be 1.00 and 0.40, respectively; the h_tot_ for soy is 7.8 mmol/g.

The in vitro protein digestibility of the soymilk gels was calculated based on measuring the pH drop after 10 min of digestion [41]. The equation was as follows: Y = 210.46 − 18.10X. Y represents the in vitro protein digestibility, and X represents the change in pH after 10 min of digestion. Three parallel tests were performed for each sample, and a pH meter (FE20, Mettler Toledo Technology (China) Co., Ltd.) was used to accurately record the decrease in pH value after 10 min (the initial pH value of each sample was adjusted to 8.0). The analysis was conducted in triplicates.

### 3.7. Peptide Content Measurement

The method outlined by Zhang et al. [42] was used to measure the content of small peptides present in samples at various digestion stages (P0, P1, P2-60, and P3-120). In brief, 2 mL of digested sample was transferred to ultrafiltration membranes (Millipore, Bedford, MA, USA) with molecular weight cut-off values of 10 kDa, and centrifuged at 8000× *g* r/min for 15 min to collect the filtrate. Fifty microliters of the filtrate were mixed with 2 mL of the pre-prepared reagent, the reaction was carried out precisely for 2 min at 25 °C, and the absorbance at 340 nm was then measured by a UV-1800 spectrophotometer (Mapada Instruments Co., Ltd., Shanghai, China). The pre-prepared reagent (50 mL) was composed of 2.5 mL of 20% (*w*/*w*) SDS, 100 μL of β-mercaptoethanol, 40 mg of OPA (dissolved in 1 mL of methanol), and 25 mL of 100 mM sodium tetraborate decahydrate. Casein tryptone (Solarbio Science and Technology Co., Ltd., Beijing, China) was used as the standard. The result was reported as the mean value of three replicates.

### 3.8. SDS-PAGE

Progressive protein degradation during GIS was analyzed using SDS-PAGE. A SDS-PAGE gel rapid preparation kit was used to prepare a 12% separating gel and a 4% stacking gel according to the manufacturer’s instructions enclosed in the kit. The digested soymilk or soymilk gel solution (50 μL) was mixed with 50 μL of loading buffer (12% (*v*/*v*) glycerol, 1.6% (*m*/*v*) SDS, 0.025% (*m*/*v*) bromophenol blue, 4% (*v*/*v*) β-mercaptoethanol, 2% (*m*/*v*) sucrose and 20% (*v*/*v*) 62.5 mM Tris-HCl, pH 6.8), and heated in boiling water for 5 min. Subsequently, 20 μL of each digested sample was added to each lane. Electrophoresis was conducted with 60 V for the stacking gel and followed by 120 V for the separating gel. The electrophoretic buffer (500 mL) was composed of 7.5 g of tris base, 36 g of glycine, and 2.5 g of SDS (dissolved in distilled water). It should be diluted 5 times before use. A pre-stained molecular mass standard (15, 20, 25, 35, 50, 70, 100, and 130 kDa, Biosharp, Labgic Technology Co., Ltd., Hefei, China) was used. The SDS-PAGE gels were stained with Coomassie Brilliant Blue R-250 after electrophoresis and analyzed using the Quantity One software (Version 4.6.2, Hercules, CA, USA).

### 3.9. Free Amino Acids Determination

As previously reported by Aro et al. [43], the concentration of free amino acids in soymilk and both soymilk gels at the end of intestinal digestion was measured using an automated amino acid analyzer HITACHI L-8900 (Hitachi Ltd., Tokyo, Japan). To precipitate large-molecular-weight proteins, samples collected at 120 min of intestinal digestion (P3-120) were mixed with 4% (*m*/*v*) trichloroacetic acid (Chinasun Specialty Products Co., Ltd., Changshu, Jiangsu, China) at a volume ratio of 1:1, and incubated at 37 °C for 30 min. Subsequently, Whatman No. 1 filter paper was used, followed by 0.45 μm aqueous membrane filtration (Jinlan Instrument Manufacturing Co., Ltd., Shanghai, China), and finally, 20 μL of the filtrate was analyzed. The result was reported as the mean value of two replicates.

### 3.10. Statistical Analysis

Microsoft Excel 2013 and SPSS 16.0 were utilized for the statistical analysis. Significant differences between means were determined by using analysis of variance and Duncan’s multiple comparison tests, and statistical significance was set at *p* < 0.05.

## 4. Conclusions

In the present study, the protein bioaccessibility of soymilk gel with the addition of GDL (SG) and soymilk gel fermented by LAB (SL) was studied by using an in vitro GIS model. The results showed that the in vitro protein digestibility of SL (80.57 ± 1.53%) was significantly higher (*p* < 0.05) than that of SG (75.33 ± 1.00%). Although similar protein degradation patterns were observed between SG and SL according to SDS-PAGE, SL digest bands had higher intensities. Moreover, the soluble protein content of SL was also much higher than that of SG during the intestinal digestion (P3) stage. In general, SL soymilk gel had a higher protein bioaccessibility than SG. This work contributes to a better understanding of the effect of acid coagulants on soy protein bioaccessibility in a simulated gastrointestinal environment. It will also be useful in giving foundational knowledge for future applications of LAB in acid-induced soymilk gel manufacturing with high nutrient-releasing characteristics. However, the gel-forming processes of SG and SF are different according to Figure 5, whether the thermal treatment temperature (42 °C or 85 °C) and time (30 min or 4 h) will affect the soy protein bioaccessibility or not are still unknown. Further research is needed to investigate the gel-forming parameters (e.g., strain type, incubation temperature, fermentation time, and ratio of bean/water) on the digestibility of acid-induced soymilk gels. Moreover, the types and magnitudes of inter- and intra-molecular forces on soy proteins in SG and SL should also be studied.

## Figures and Tables

**Figure 1 molecules-27-06202-f001:**
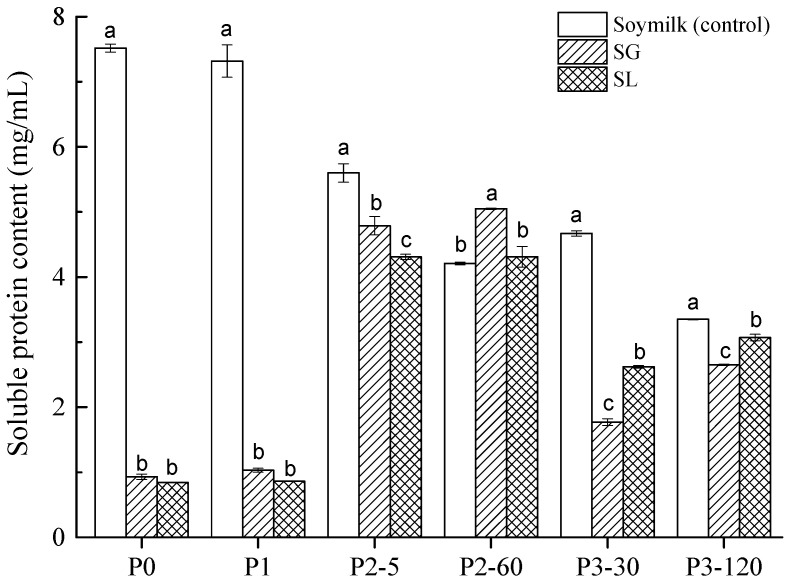
Soluble protein content (mg/mL) of soymilk (control), soymilk gel produced by GDL (SG), and soymilk gel induced by LAB (SL) samples at various digestion stages. Within the same digestion phase, different lowercase letters indicate significant differences (*p* < 0.05). P0 represents the sample before in vitro GIS digestion; P1 represents the sample after buccal digestion; P2-5 and P2-60 indicate samples collected at 5 min and 60 min of gastric digestion, respectively; and P3-30 and P3-120 indicate samples collected at 30 min and 120 min of intestinal digestion, respectively.

**Figure 2 molecules-27-06202-f002:**
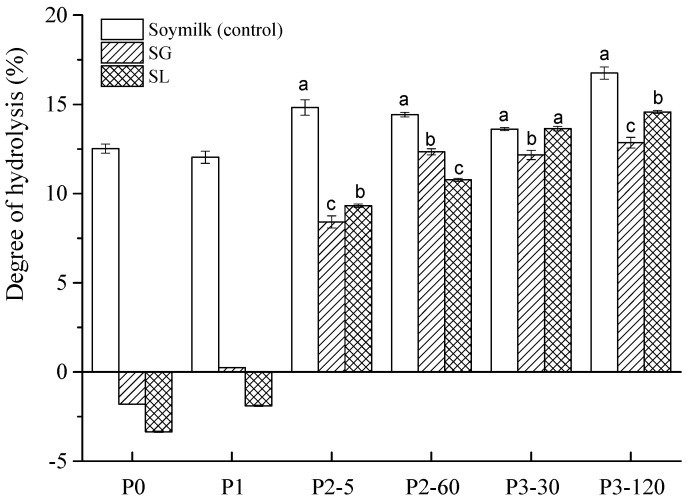
Degree of hydrolysis (DH) of soymilk (control), soymilk gel produced by GDL (SG), and soymilk gel induced by LAB (SL) samples subjected to in vitro GIS. Within the same digestion phase, different lowercase letters indicate significant differences (*p* < 0.05). P0 represents the sample before in vitro GIS digestion; P1 represents the sample after buccal digestion; P2-5 and P2-60 indicate samples collected at 5 min and 60 min of gastric digestion, respectively; and P3-30 and P3-120 indicate samples collected at 30 min and 120 min of intestinal digestion, respectively.

**Figure 3 molecules-27-06202-f003:**
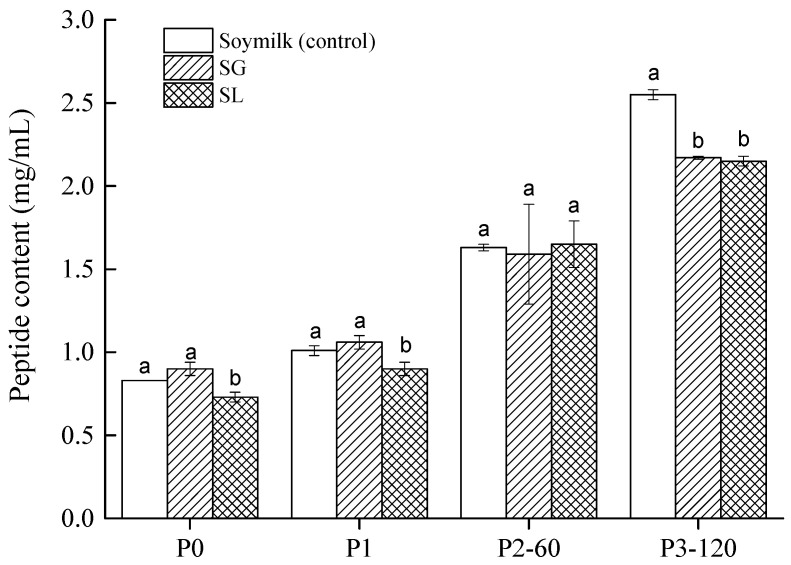
Peptide contents (<10 kDa) in soymilk (control), soymilk gel produced by GDL (SG), and soymilk gel induced by LAB (SL) samples. Within the same digestion phase, different lowercase letters indicate significant differences (*p* < 0.05). P0 represents the sample before in vitro GIS digestion; P1 represents the sample after buccal digestion; P2-5 and P2-60 indicate samples collected at 5 min and 60 min of gastric digestion, respectively; and P3-30 and P3-120 indicate samples collected at 30 min and 120 min of intestinal digestion, respectively.

**Figure 4 molecules-27-06202-f004:**
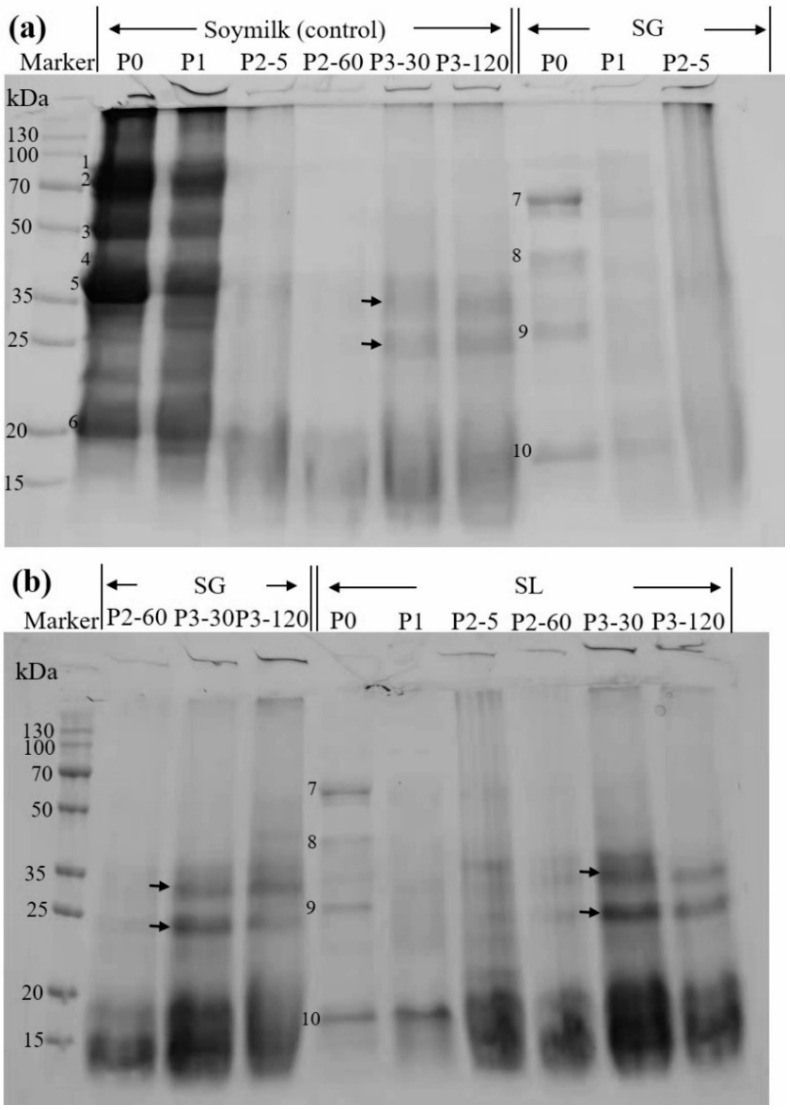
SDS-PAGE profiles of proteins that were taken from the supernatant at various stages of in vitro GIS. Predominant bands seen both before and after in vitro GIS were denoted by numbers 1–10. (**a**) Soymilk and SG digesta, (**b**) SL digesta. P0 represents the sample before in vitro GIS digestion; P1 represents the sample after buccal digestion; P2-5 and P2-60 indicate samples collected at 5 min and 60 min of gastric digestion, respectively; and P3-30 and P3-120 indicate samples collected at 30 min and 120 min of intestinal digestion, respectively.

**Figure 5 molecules-27-06202-f005:**
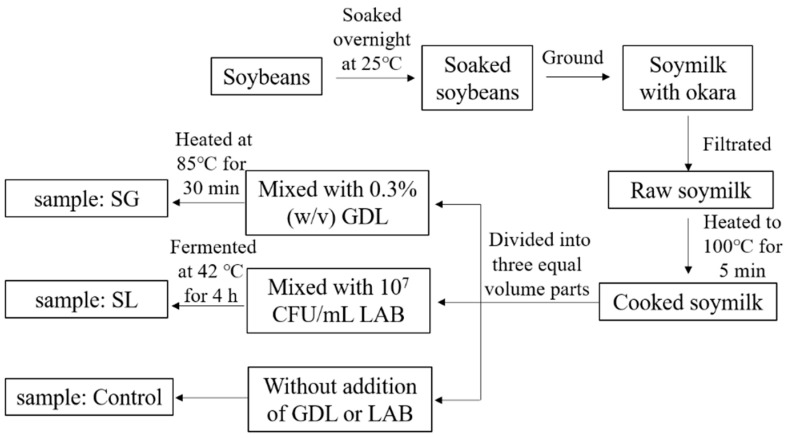
Schematic diagram of the step-by-step preparation of soymilk gels.

**Figure 6 molecules-27-06202-f006:**
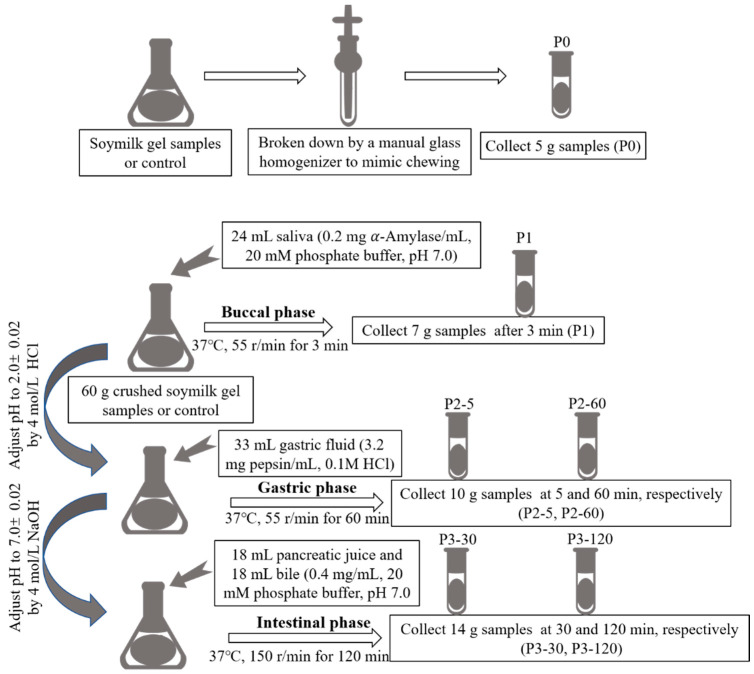
The whole in vitro GIS digestion steps.

**Table 1 molecules-27-06202-t001:** In vitro digestibility of soymilk and both soymilk gels ^1,2^.

Sample	In Vitro Digestibility (%)
Soymilk (control)	82.16 ± 1.12 ^a^
SG	75.33 ± 1.00 ^c^
SL	80.57 ± 1.53 ^b^

Note: ^1^ Values are represented as means ± standard deviations. ^2^ Mean values with a distinct letter (a–c) are significantly different at the significance (*p* < 0.05) level.

**Table 2 molecules-27-06202-t002:** Free levels of soymilk and both soymilk gels at the end of in vitro digestion. The results are presented in grams per liter of digestion solution.

Amino Acid Types	Soymilk (Control)	SG	SL
Asp	0.160	0.140	0.146
Thr	0.090	0.070	0.083
Ser	0.121	0.125	0.127
Glu	0.823	0.667	0.824
Gly	0.135	0.091	0.104
Ala	0.622	0.463	0.491
Cys	0.026	0.011	0.030
Val	0.074	0.067	0.070
Met	0.005	0.003	0.004
Ile	0.000	0.000	0.002
Leu	2.753	2.523	2.476
Tyr	5.077	4.578	4.638
Phe	1.788	1.731	1.764
Lys	4.749	5.483	5.550
His	0.280	0.287	0.278
Trp	4.040	3.528	3.961
Pro	0.111	0.084	0.089
Essential amino acids	13.499	13.405	13.910
Total	20.854	19.851	20.637

## Data Availability

Not applicable.

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
