# Peer review of "Comparison on Protein Bioaccessibility of Soymilk Gels Induced by Glucono-δ-Lactone and Lactic Acid Bacteria"

_molecules, 2022, doi:10.3390/molecules27196202_

Round 1
Reviewer 1 Report
The manuscript evaluated the protein bioaccessibility of soymilk gels produced by the addition of glucono-lactone (GDL) and fermentation with lactic acid bacteria (LAB). However, there has a lot of problems. 1. In Introduction. the specific advantages of lactic acid bacteria (LAB) and the influence mechanism of metabolites and properties of LAB (polysaccharides, proteases, etc) on gel produced by LAB should be briefly explained. 2. “2.1 Proximate analysis” should examine the chemical compositions of soybean gel. 3.In Figure 2. Negative degree of hydrolysis?4. The discussion should be in-depth and comprehensive, especially comparative analysis with the latest literature.
5、The English grammar needs to be corrected.
Reviewer 2 Report
The article is well written and well structured. It lacks interest and does not provide new knowledge, there is not contribution. Main results have been published since 2016. "A comparison study of bioaccessibility of soy protein gel induced by magnesiumchloride, glucono-d-lactone and microbial transglutaminase. DOI:10.1016/j.lwt.2016.03.032. And "Differences in the physicochemical, digestion and microstructural characteristics of soy protein gel acidified with lactic acid bacteria, glucono-d-lactone and organic acid. https://doi.org/10.1016/j.ijbiomac.2021.06.071.
Last article has a similar discussion to this submission.
The only new part is the amino acid profile, but the author does not focus his article on that results.
Author Response
Thank you very much for all your valuable suggestions. We really appreciate your approval on the structure of our manuscript. Both of the articles mentioned by the reviewer#2 are excellent works, which reported different coagulants had vital influence on the protein bioaccessibility of proteins in soymilk curds. However, the authors of these two articles did not set soy protein isolate as the control (the raw material). It’s important to compare the soy protein gels with the raw material to understand how coagulants affect the digestibility of soy proteins. Therefore, in the present study, we set the cooked soymilk as control, and give a detailed comparison on the protein bioaccessibility of soymilk gels induced by GDL and LAB. It will help provide fundamental knowledge for future applications of LAB in the acid-induced soymilk gel manufacturing industry with high nutrient-releasing properties.
Reviewer 3 Report
Dear author,
This current manuscript investigated protein bioaccessibility of two kind of soymilk gels induced by GDL and LAB. Overall, a very detailed paper with excellent figures representing the significant results. I have some suggestions and questions below,
1. improve the English language. some expression are easily confused, such as line 76, 81, 95, 127, 134, 155. 179, Please recheck and correct it accordingly.
2. This paper mainly investigated soy gels induced by GDL and LAB, when comparing these process of gel-forming, thermal treating and acid effect were total different, however, litter discussion was mentioned in the discussion. it is strongly recommended these contents should be considered before make conclusion.
3. SDS-PAGE loading amount should be relatively consistent. Otherwise it is difficult to compare. At present, the relative volume of sample is sure according to the methods. However, the protein concentration of sample is still unclear. in addition, there is obvious difference in loading amount of protein of SDS-PAGE, please explain.
4. line 239-241,there is no statistical result. In addition, there is no difference in the data between the groups. Therefore, the conclusion is worth reconsidering.
5. The conclusion is not insightful, Please provide more suggestions.
Author Response
We thank the reviewers for their valuable comments and suggestions on our manuscript sincerely. We have revised the manuscript accordingly, and included a detailed list of responses below. The according changes mentioned here have been highlighted in red fonts in the revised manuscript.
Question 1: This current manuscript investigated protein bioaccessibility of two kind of soymilk gels induced by GDL and LAB. Overall, a very detailed paper with excellent figures representing the significant results. I have some suggestions and questions below.
improve the English language. some expression are easily confused, such as line 76, 81, 95, 127, 134, 155. 179, Please recheck and correct it accordingly.
Response:
Thank you very much for all your valuable suggestions. We really appreciate your approval on this manuscript. As suggested, we have changed the expression mentioned above. Please refer to Lines 91, 96, 112, 150, 157, 188 and 212 in the revised manuscript to see the changes.
Question 2: Results and Discussion
This paper mainly investigated soy gels induced by GDL and LAB, when comparing these process of gel-forming, thermal treating and acid effect were total different, however, litter discussion was mentioned in the discussion. it is strongly recommended these contents should be considered before make conclusion.
Response:
We strongly agree with the reviewer’s comments. We add some discussion in the Conclusion section as follows:
However, the gel-forming process of SG and SF are different according to Figure 5, whether the thermal treatment temperature (42℃ or 85℃) and time (30 min or 4 h) will affect the soy protein bioaccesibility are still unknown. Further research is needed to investigate the gel-forming parameters (e.g., strain type, incubation temperature, fer-mentation time, and ratio of bean/water) on the digestibility of acid-induced soymilk gels.
Please refer to Lines 446-451 in the revised manuscript to see the changes.
Question 3: SDS-PAGE loading amount should be relatively consistent. Otherwise it is difficult to compare. At present, the relative volume of sample is sure according to the methods. However, the protein concentration of sample is still unclear. in addition, there is obvious difference in loading amount of protein of SDS-PAGE, please explain.
Response:
The soluble protein concentration of soymilk and soymilk gel samples at different digestion time points is shown in Figure 1. The digested soymilk or soymilk gel solution (50 μL) was mixed with 50 μL loading buffer, and heated in boiling water for 5 min. Subsequently, 20 μL of each digested sample was added to each lane. As the soluble protein content was different in each sample, the intensity of protein bands was different as shown in Figure 4.
Question 4: line 239-241,there is no statistical result. In addition, there is no difference in the data between the groups. Therefore, the conclusion is worth reconsidering.
Response:
The conclusion here has been corrected, please refer to Lines 280-281 in the revised manuscript to see the changes.
Question 5: The conclusion is not insightful, Please provide more suggestions.
Response:
We provide more suggestions in the conclusion section, please refer to Lines 446-452 in the revised manuscript to see the changes.
Round 2
Reviewer 1 Report
The manuscript has been revised as required
Author Response
Comments: The manuscript has been revised as required.
Response:
It is a great honor to receive your recognition for this work. Your comments have greatly improved the quality of this manuscript, thanks very much.
Reviewer 2 Report
Conclusion:
Line 461-462: “It suggested that SL was more susceptible to digestive enzymes than SG”.
line 473: “Collectively, SL 473 soymilk gel had a higher protein bioaccessibility than SG”.
These are all the conclusion of the article. Author did not improve article to show new scientific contribution in the area.
Author Response
Comments:
Conclusion: Line 461-462: “It suggested that SL was more susceptible to digestive enzymes than SG”. line 473: “Collectively, SL 473 soymilk gel had a higher protein bioaccessibility than SG”. These are all the conclusion of the article. Author did not improve article to show new scientific contribution in the area.
Response:
We are very grateful for your comments and thoughtful suggestions. Now, in the revised manuscript, we have deleted the sentence "It suggested that SL was more susceptible to digestive enzymes than SG" to avoid repetition. We have also corrected the Conclusion section by adding the following description to show new scientific contribution in the area:
This work contributes to a better understanding of the effect of acid coagulants on soy protein bioaccessibility in a simulated gastrointestinal environment. It will also be useful in giving foundational knowledge for future applications of LAB in acid-induced soymilk gel manufacturing with high nutrient-releasing characteristics.
Please refer to lines 444-448 in the revised manuscript to see the changes.
Reviewer 3 Report
The author has carefully revised and replied to the questions raised. At present, there are no further suggestions for revision. It is recommended to accept
Author Response
Comments: The author has carefully revised and replied to the questions raised. At present, there are no further suggestions for revision. It is recommended to accept.
Response: We feel great thanks for your professional review work on our article. Thank you again for your positive comments and valuable suggestions to improve the quality of our manuscript.